# Structure, Activity, and Function of PRMT1

**DOI:** 10.3390/life11111147

**Published:** 2021-10-27

**Authors:** Charlène Thiebaut, Louisane Eve, Coralie Poulard, Muriel Le Romancer

**Affiliations:** 1Université de Lyon, F-69000 Lyon, France; charlene.thiebaut@inserm.fr (T.C.); louisane.eve@lyon.unicancer.fr (E.L.); coralie.poulard@lyon.unicancer.fr (P.C.); 2Inserm U1052, Centre de Recherche en Cancérologie de Lyon, F-69000 Lyon, France; 3CNRS UMR5286, Centre de Recherche en Cancérologie de Lyon, F-69000 Lyon, France

**Keywords:** PRMT1, arginine methylation, H4R3 methylation, transcriptional regulation, cell signaling, DNA damage repair, cancer

## Abstract

PRMT1, the major protein arginine methyltransferase in mammals, catalyzes monomethylation and asymmetric dimethylation of arginine side chains in proteins. Initially described as a regulator of chromatin dynamics through the methylation of histone H4 at arginine 3 (H4R3), numerous non-histone substrates have since been identified. The variety of these substrates underlines the essential role played by PRMT1 in a large number of biological processes such as transcriptional regulation, signal transduction or DNA repair. This review will provide an overview of the structural, biochemical and cellular features of PRMT1. After a description of the genomic organization and protein structure of PRMT1, special consideration was given to the regulation of PRMT1 enzymatic activity. Finally, we discuss the involvement of PRMT1 in embryonic development, DNA damage repair, as well as its participation in the initiation and progression of several types of cancers.

## 1. Introduction

Arginine methylation is a common and widespread post-translational modification (PTM) in eukaryotes that regulates numerous biological processes. Currently, nine protein arginine methyltransferases (PRMTs) have been described which are divided into three families according to the type of methylarginine produced. Type I PRMTs (PRMT-1, 2, 3, 4, 6 and 8) generate ω-N^G^-monomethylarginine (MMA) and ω-N^G^, N^G^-asymmetric dimethylarginine (ADMA), Type II PRMTs (PRMT-5 and 9) generate MMA and ω-N^G^, N’^G^-symmetric dimethylarginine (SDMA) and finally the unique Type III PRMT, PRMT7, generates MMA. Mechanistically, all PRMTs catalyze the transfer of a methyl group from S-adenosyl methionine (AdoMet) to the guanidino nitrogen atom of arginine [1]. Though considered for a long time as a stable mark, it is now well-known that arginine methylation is a dynamic PTM that can be removed by arginine demethylases [2].

PRMT1, which is the major type I PRMT, is responsible for 85% of the activity attributed to type I PRMTs in mammals [3]. Moreover, it plays key roles in various cellular processes such as transcriptional regulation, signal transduction or DNA damage repair, owing to the diversity of its histone and non-histone substrates [1].

The aim of this review is to provide an overview of the literature concerning PRMT1 structure, activities and functions. After a detailed description of the genomic organization and the protein structures of the different PRMT1 isoforms, the substrate specificity and the regulatory mechanisms of PRMT1 itself will be discussed. Finally, the cellular roles and functions of PRMT1, as well as its involvement in cancer, will be addressed. 

## 2. Structural Features

### 2.1. Genomic Organization

Human PRMT1 is encoded by the *PRMT1* gene located on chromosome 19 (19q13.3) and composed of 12 exons and 11 introns. At the 5′ end of this genomic locus of 11.3 kilobases (kb) are four alternative exons (e1a-e1d) involved in the synthesis of at least seven splice variants of PRMT1 (v1–v7) [4,5] (Figure 1A,B). More recently, next-generation sequencing led to the identification of a novel exon located between exons 11 and 12, and 58 additional alternative splice variants of the *PRMT1* gene. Among them, 34 are speculated to encode additional protein isoforms of PRMT1 but remain to be characterized [6]. 

### 2.2. Protein Structure

At the protein level, human PRMT1 shares a high degree of homology with the different members of the PRMT family that is conserved in eukaryotes. Phylogenetic studies based on the methyltransferase domain highlighted that PRMT1 is closely related to PRMT8 [7]. The canonical structure of PRMT1 includes three functional domains: (i) the N-terminal methyltransferase domain characterized by the Rossmann fold constituting the AdoMet binding pocket, (ii) the C-terminal β-barrel domain which forms a cylindrical structure corresponding to the arginine-substrate binding site and (iii) the α-helical dimerization arm which originates from the N-terminal part of β-barrel domain and connects to the Rossmann fold of a second monomer [8]. 

The catalytic core of PRMT1 is composed of 6 highly conserved peptide motifs essential for the methyltransferase activity. Motif I (VLDVGSGTG) delimits the AdoMet-binding site and is stabilized by motifs II (VDI) and III (LAPDG). The binding of the AdoMet in this pocket is favored by the formation of hydrogen bonds with the glutamic acid residue of the post-motif I (VIGIE). In addition, the double-E motif (SEWMGYCLFYESM) and the THW loop (YTHWK) define the peptidyl arginine-substrate pocket (Figure 1C). The double-E motif is composed of two negatively charged glutamic acid residues (E144 and E153) that neutralize the positively charged guanidium group of the target arginine, whereas the THW loop stabilizes three dynamic α-helices (αX, αY, αZ) located at the N-terminus of the Rossmann fold that participates in peptidyl arginine recognition [9,10,11]. To illustrate the organization of the catalytic core of PRMT1, an extensive study of the crystal structure of rat PRMT1 which shares 96% identity with the amino acid sequence of human PRMT1 was performed by Zhang and Cheng [10].

Dimerization of PRMTs is a conserved process, crucial for their methyltransferase activity. This mechanism is mediated by the dimerization arm that interacts with the outer surface of the AdoMet binding site through hydrophobic interactions and hydrogen bonds [12]. PRMT1 mutants displaying a mutation or a deletion of the dimerization arm were key to demonstrating the importance of dimerization for AdoMet binding, substrate specificity and the processivity of the methyltransferase activity [10]. As previously described for the yeast PRMT1 counterpart, Hmt1, rat PRMT1 dimers can be assembled into oligomers through hydrophilic interactions [13,14]. This oligomerization is notably associated with a stimulation of the PRMT1 methyltransferase activity [14].

### 2.3. PRMT1 Isoforms

To date, seven PRMT1 isoforms, PRMT1-v1 to PRMT1-v7, that differ in length and sequence of their N-terminal region have been identified (Figure 1B). These variations of the N-terminal sequence can impact enzymatic activity and substrate specificity. Unlike PRMT1-v7 which is catalytically inactive, variants PRMT1-v1 to PRMT1-v6 exhibit a methyltransferase activity in vitro on different previously described PRMT1 substrates. However, PRMT1-v3 and PRMT1-v4 display a lower methylation efficiency compared to the others. Studies of Goulet et al. also showed that each substrate can be preferentially methylated by a particular isoform. For example, Sam68 and SmB are mainly methylated by PRMT1-v1 and PRMT1-v2 [5]. Currently, studies describing the functionality of the PRMT1-v7 variant are lacking. Although it has retained the ability to heterodimerize with other isoforms, it does not seem to be involved in the regulation of their activity [5].

Differences in enzymatic activities observed among the different PRMT1 isoforms can be partly explained by their subcellular localization. Using a GFP-PRMT1 isoform reporter system, Goulet et al. showed that PRMT1-v1 and -v7 are mainly nuclear, whereas PRMT1-v2 is primarily cytoplasmic [5]. The nucleocytoplasmic shuttling of PRMT1-v2 depends on a leucin-rich nuclear export sequence (NES) encoded by the retained exon 2, but also on its enzymatic activity [15]. Interestingly, there is also a tissue-specific expression pattern of the different PRMT1 isoforms. PRMT1-v1, -v2 and -v3 are ubiquitously expressed in human tissues [4], whereas PRMT1-v4 to -v7 are tissue-specific. More precisely, expression of PRMT1-v4 and -v5 is restricted to the heart and pancreas, respectively; yet, PRMT1-v7 is detectable in the heart and skeletal muscle. PRMT1-v6 expression has so far not been detected in any normal human tissues but was detected in certain breast cancer cell lines [5]. 

## 3. Biochemical Features

### 3.1. Sequence Specificity

PRMT1, like the other type I PRMTs, except PRMT4, catalyzes the asymmetric dimethylation of arginine residues localized in glycine/arginine rich regions and more particularly within RGG or RXR motifs [10]. “RGG” sequences that are often located in regions rich in “RG” dinucleotides are also described as “RGG/RG” motifs that can be subdivided into 4 categories according to the number of repeats: “Tri-RGG”, “Di-RGG”, “Tri-RG” or “Di-RG” motifs [16]. Many substrates of PRMT1 contain a combination of these different motifs such as TAF15 (3 Tri-RGG, 1 Di-RGG) or Sam68 (1 Di-RGG, 1 Tri-RG, 1 Di-RG). Structurally, the presence of glycine residues near the target arginine induces a conformational flexibility that facilitates substrate recognition [17]. 

The modification of a single residue in conserved motifs like “RGG” can abolish the activity of PRMT1 towards the mutated substrate. For instance, the helicase eIF4A1 that contains an “RGG” motif is methylated by PRMT1, whereas the eIF4A3 isoform in which “RGG” is replaced by an “RSG” sequence is not a substrate for PRMT1. However, it was shown in the same study that PRMT1 is able to methylate synthetic peptides that contain a “RSG” sequence [18]. This suggests that other residues located at a long distance from the target arginine can also be involved in its recognition. This hypothesis was substantiated by a study of Osborne et al., which showed that the affinity of PRMT1 for its arginine substrate relies on long-range interactions involving an acidic residue located away from the PRMT1 active site and probably a positively-charged residue on the substrate [19].

### 3.2. Product Specificity

Understanding mechanisms that regulate the degree (mono- or dimethylation) and the type (symmetric or asymmetric) of methylation catalyzed by each member of the PRMT family is a major challenge. Indeed, MMA, ADMA and SDMA induce distinct and sometimes antagonistic biological effects as notably described for mono- and dimethylated H3R2 [20,21]. 

Studies conducted by Gui et al. on rat PRMT1 that shares 96% sequence identity with its human counterpart, identified two conserved methionine residues, M48 and M155, located in the active site that position the target arginine in a favorable configuration for asymmetric dimethylation. Interestingly, M48 also participates in the specific recognition of the target arginine in multi-arginine protein substrates [20]. Mutations in M48L and M155A induce an imbalance in the proportion of MMAs and ADMAs, but do not allow SDMA generation [20]. However, when M48 is mutated to phenylalanine (M48F), a switch in PRMT1 activity occurs, enabling it to induce symmetrical dimethylation. This is consistent with the fact that product specificity of PRMT5 which catalyzes SDMA formation is controlled by the conserved F379 residue in its active site [22]. More recently, mutagenesis studies showed that H293S mutation of the PRMT1 active site does not affect the production of MMA and ADMA by itself, but leads to a predominant formation of SDMA when it is associated with the M48F mutation [23]. 

The product specificity of PRMT1 which is non-stochastic and regioselective can also be guided by the substrate itself. It seems that the N-terminal arginyl-groups of substrates constitute the main targets for PRMT1 methylation, whereas positively-charged C-terminal residues (including arginines) participate in long-range interactions with acidic residues of PRMT1. This strengthens the affinity of PRMT1 for its arginine substrates [19,24]. 

Interestingly, the amino acid sequence of the substrate can also direct the degree of methylation (mono- or dimethylation) by regulating PRMT1 processivity [24,25]. Whether PRMT1 dimethylates its substrates in a distributive or processive manner is a matter of debate in the literature. While numerous studies support that PRMT1 acts distributively by transiently releasing MMA and replacing the methyl donor between the two methyl-group transfers [26,27,28], Obianyo and co-workers described a semi-processive activity of PRMT1. In this model the mono-methylated intermediate remained associated with the enzyme but the product S-adenosylhomocysteine (AdoHcy) was replaced by a novel AdoMet to allow the second reaction [19,29,30]. Studies on the catalytic activity and processivity of PRMT1 are ongoing, and the latest data indicate that the degree of processivity of PRMT1 depends on its dimerization but is also dependent on cofactor or enzyme concentrations [10,25].

### 3.3. Regulation of PRMT1 Expression and Enzymatic Activity

Many studies have sought to decipher the different levels of regulation of PRMT1 expression and enzymatic activity. Indeed, substrate methylation by PRMT1 is a highly regulated and dynamic phenomenon, occurring directly through PRMT1 PTMs or through its association with co-regulators. In addition, crosstalk between different PTMs on the same substrate can influence arginine methylation by PRMT1. Finally, methyl marks on arginine can be removed by PAD4 which demethylates histones by converting MMA to citrulline [31] or by JMJD6 which directly removes the methyl group to convert methylarginine into arginine [32]. More recently, JMJD1B, a well-known lysine demethylase for H3K9me2, has also been described as effective in demethylating H4R3me1 and H4R3me2a [33]. 

#### 3.3.1. Regulation of PRMT1 Expression

PRMT1 can be regulated at the level of its expression. Indeed, a very recent study discovered that the serine/threonine kinase mTOR is involved in the regulation of PRMT1 expression in a fasting context. Forty-eight hours of experimental fasting was shown to induce a decrease in STAT1 phosphorylation mediated by mTOR, leading to the inhibition of STAT1 binding to the PRMT1 promoter. In this fasting condition, the decrease in PRMT1 expression induced a decrease in mitochondrial mass and thus a decrease in cellular energy availability [34]. Moreover, the expression level of PRMT1 can also be regulated by micro-RNAs (miR). This is the case for example for miR-503 that has a tumor suppressor role and whose expression is low in several types of cancers. In hepatocellular carcinoma cells, miR-503 directly targets PRMT1 and reduces its expression level. Consequently a decrease in cell invasion, migration and epithelial-mesenchymal transition are observed [35].

#### 3.3.2. Post-Translational Modification of PRMT1

Unlike other PRMTs, few PTMs of PRMT1 have been described to date. A first study in 2004 conducted using mass spectrometry found that PRMT1 is phosphorylated on Y291. Using non-natural amino acid mutagenesis, the authors showed that phosphorylation of PRMT1 on Y291 alters protein-protein interactions and substrate specificity. Indeed, Y291 phosphorylation of PRMT1 decreases its interaction with hnRNP, and enzymatic activity on hnRNP in vitro. This is due to the negative charge of the phosphate group that modifies the tertiary structure of the enzyme and in particular of the THW loop [36]. Following this first finding, another study in keratinocytes revealed that PRMT1 is a substrate of the kinase CSNK1a1. Although phosphorylation of PRMT1 by CSNK1a1 does not affect the methylation efficiency of PRMT1 on several known substrates, it seems that it modulates its transcriptional activity on some target genes. Indeed, phosphorylated PRMT1 seems to induce the transcription of genes involved in proliferation and repress the expression of genes involved in keratinocyte differentiation [37]. More recently, in ovarian cancer cells, it was shown that PRMT1 can be phosphorylated by DNA-PK in response to cisplatin, thus inducing its recruitment on chromatin and its enzymatic activity towards H4R3 [38]. 

PRMT1 activity is also modulated by its degradation mediated by the proteasome pathway. In this context, a study in human embryonic kidney cells showed that PRMT1 is polyubiquitinated by the E3 ubiquitin ligase, TRIM48. Thus, the polyubiquitination of PRMT1 decreases the level of methylation of the substrate ASK1, a kinase involved in the cellular stress response. Downregulation of PRMT1 thus promotes cell death induced by ASK1-mediated oxidative stress. Polyubiquination of PRMT1 also negatively impacts FOXO1 methylation and its transcriptional activity [39]. Another in vivo study used an engineered ubiquitin transfer method called “orthogonal UB transfer” to profile E3 substrate specificity. This method showed that PRMT1 is polyubiquitinated by two other E3 ubiquitin ligases, CHIP and E4B, leading to its proteasome-mediated degradation. Nevertheless, the physiological consequences of this polyubiquitination were not investigated in this study [40]. Given the importance of PRMT1, it probably undergoes many other PTMs including methylation, such as PRMT5 which is methylated by PRMT4 [41], or OGT-glycosylation [42]. Although other modifications (i.e., acetylation and sumoylation) have not been described in the PRMT family, it is likely that these events exist.

#### 3.3.3. PRMT1 Association with Co-Regulators

PRMT1 activity can also be regulated through its interaction with non-substrate proteins that modulate its methyltransferase activity. The first regulators were described in 1996, with the BTG1 (B-cell translocation gene 1) and BTG2. This study showed in vitro that the interaction of BTG1 and BTG2 with PRMT1 positively modulates its enzymatic activity towards a substrate, hnRNPA1 [43]. Several years later, our team discovered a new regulator of PRMT1, hCAF1. We showed by in vitro methylation assay that hCAF1 inhibits PRMT1-mediated methylation of histone H4 on arginine 3 (H4R3) by PRMT1. This observation was confirmed in breast cancer cells where depletion of hCAF1 induces a strong reduction in the overall level of asymmetric arginine methylation, indicating that hCAF1 modulates PRMT1 activity towards several substrates [44]. Interestingly, a study in HeLa cells revealed a crosstalk between PRMT1 and PRMT2. Indeed, PRMT2 binds to PRMT1 without methylating it and potentiates its enzymatic activity towards H4R3. Surprisingly, PRMT2-mediated activation of PRMT1 also induces an increase in SDMA levels in vivo, implying possible further crosstalk between the different enzymes of the PRMT family [45].

PRMT1 activity can also be modulated by exogenous regulators. For instance, the serine/threonine phosphatase PP2A has been described to regulate PRMT1 activity. PRMT1 methylate hepatitis C virus NS3 protein and inhibits its helicase activity. PP2A binds to PRMT1 and inhibits its enzymatic activity towards a NS3 protein, which affects inhibitory role of PRMT1 on the helicase activity of NS3. Interestingly, the hepatitis C virus upregulates PP2A expression, thus counteracting the downregulation of NS3 by PRMT1. This study highlights the complexity of the pathways regulating PRMT1 enzymatic activity [46].

In addition, other regulators have been identified, such as RALY [47], TR3 [48], PDGF-BB [49], or GFI1 [50]. Moreover, other mechanisms of regulation of PRMT1 have been uncovered, such as oxidative stress [12] or iron deficiency [51].

#### 3.3.4. PTMs Influencing PRMT1 Activity

In parallel to the direct regulation of PRMT1 by PTMs or by the binding of co-regulators, a crosstalk between arginine methylation and different PTMs deposited by other enzymes on the same substrate has been described. For example, a 2006 study showed that methylation of H4R3 by PRMT1 at the pS2 promoter is required to activate its expression. Interestingly, this study showed that histone hypoacetylation is necessary for the recruitment of PRMT1 to the promoter and for the deposition of the H4R3 methylation mark. The patient SE translocation (SET) protein, which is part of the INHAT complex, prevents the acetylation of the histone at the pS2 promoter [52]. Another study investigated the effect of histone H4 phosphorylation on serine 1 (H4S1). The authors showed by in vitro methylation assays that H4S1 phosphorylation leads to a 3-fold decrease in PRMT1-mediated H4R3 methylation. Interestingly, mass spectrometry analysis revealed MMA as a PRMT1 major product. Indeed, further in vitro methylation assays revealed a 3-fold decrease in ADMA, due to an approximate 11-fold reduction in PRMT1 catalytic efficiency. Moreover, H4S1 phosphorylation also leads to a 8-, 5-, and 3-fold decrease in PRMT3, PRMT8 and PRMT5 activity, respectively [53]. 

These in vitro studies highlighted the complex crosstalk between the different PTMs in the histone code and the tight regulation of the activity of each enzyme. Although this phenomenon has only been described on H4R3 for PRMT1, this is probably because PRMT1 was first described as a histone methyltransferase catalyzing H4R3 methylation [54]. Many non-histone substrates have since been described, and likely display similar crosstalk that remains to be depicted.

### 3.4. Substrates

Arginine 3 of histone H4 was the first substrate described for PRMT1 [54,55]. The asymmetric dimethylation of H4R3 constitutes an activating mark of transcription [56]. It was also demonstrated that PRMT1 methylates histone H2A at R3, R11 and R29, although the latter two are not localized within a consensus motif recognized by PRMT1 [57]. Further studies are expected to clarify the impact of these two histone marks on transcriptional activity. In addition to the activity of PRMT1 as a chromatin modifying enzyme, a plethora of non-histone substrates of PRMT1 have been identified and can be classified according to their cellular functions: transcriptional and translational regulation, RNA-processing, DNA damage repair and signal transduction. A list of the currently identified substrates of PRMT1 is available in Table 1.

It is important to note that some substrates are common to different types of PRMTs and that competitive mechanisms may exist. This hypothesis is supported by the observations of Dhar et al. who showed that inhibition of PRMT1 induces a decrease in the level of ADMA concomitant with an increase in MMA and SDMA levels [58].
life-11-01147-t001_Table 1Table 1List of non-histone substrates of PRMT1 classified according to their cellular functions.Biological FunctionSubstrateMethylation SiteBiological OutcomeReferenceTranscriptionalRegulationTranscriptionalregulationBRCA1Within the 504–802 regionPromotes BRCA1 recruitment to specific promoters[59]C/EBPαR35, R156, R165Prevents C/EBPα interaction with the corepressor HDAC3[60]c-MycR299, R346Promotes c-Myc interaction with p300[61]EZH2R342Prevents EZH2 target gene expression[62]FOXO1R248, R250Prevents FOXO1 phosphorylation by Akt[63]FOXP3R48, R51Enhances FOXP3 transcriptional activity[64]GLI1R597Enhances GLI1 binding to target gene promoters[65]MyoDR121Promotes MyoD DNA-binding and transcriptional activity[66]Nrf2R437Promotes Nrf2 DNA-binding and transcriptional activity[67]PRR637Accelerates PR recycling and transcriptional activity[68]RACO-1R98, R109Promotes c-Jun/AP1 activation[69]RelAR30Prevents RelA DNA-binding and represses NF-κB target genes[70]RIP40R240, R650, R948Favors RIP140 nuclear export and prevents the recruitment of HDAC3[71]RunX1R206, R210Prevents Sin3a binding and promotes RUNX1 transcriptional activity[72]STAT1R31Prevents STAT1 association with PIAS1 and enhances IFNα/β induced transcription[73]TAF15R203Affects the subcellular localization of TAF-15 and enhances its transcriptional activity[74]FUS/TLSR216, R218, R242, R394Participates in the nuclear cytoplasmic shuttling of FUS/TLS and enhances its transcriptional activity[75,76]TOP3BR833, R835Promotes TOP3B interaction with TDRD3, stress granule localization and topoisomerase activity[77]Twist1R34Regulates the nuclear import of Twist1 and represses E-cadherin expression[78]RNA- processingCNBPR25, R27Prevents its RNA binding activity[79]G3BP1R435, R447Prevents stress granule formation during oxidative stress[80]hnRNPA1R214, R226, R223, R240Prevents hnRNPA1 ITAF activity and RNA-binding ability[81]HSP70R416, R447Enhances HSP70 RNA-binding and -stabilization abilities[82]NS3R1493Affects NS3 RNA-binding and helicase activity[46,83]RBM15R578Promotes RBM15 degradation by CNOT4 (RNA splicing)[84]Sam68Within the 276–343 regionPrevents Sam68 poly(U) RNA-binding activity[85,86]SF2/ASFR93, R97, R109Affects SF2/ASF nucleocytoplasmic distribution and modulates the alternative splicing of target genes[87,88]TranslationalRegulationeIF4A1R362Prevents eIF4A1 interaction with eIF4G1 and inhibits ATPase activity[18,89]eIF4G1R689, R698Regulates eIF4G1 stability and the assembly of the translation initiation complex[90]rpS3R64, R65, R67Promotes rpS3 import into the nucleolus and ribosome assembly[91]DNA damagerepair53BP1Within the 1319–1480 regionPromotes 53BP1 recruitment to DNA-damage sites[92]APE1R301Promotes APE1 mitochondrial translocation (translocase Tom20) and protects mitochondrial DNA from oxidative damage[93]DNA pol βR137Prevents DNA pol β interaction with PCNA in BER pathway[94]E2F-1R109Promotes E2F-1-dependent apoptosis in DNA-damaged cells[95]FEN1Not determinedStabilizes FEN1 and upregulates its DNA damage repair activities[96]hnRNPKR296, R299Prevents PKCδ-dependent apoptosis during DNA damage[97]hnRNPUL1R584, R618, R620, R645, R656Promotes hnRNPUL1 association with NBS1 and recruitment to DNA-damage sites[98]MRE11GAR domainPromotes MRE11 recruitment to DNA-damage sites and favors its exonuclease activity[99,100]RunX1R233, R237Confers resistance to apoptosis under stress condition and DNA damage accumulation[101]ASK1R78, R80Prevents the stress-induced ASK1-JNK1 signaling[102]Signal transductionAxinR378Favors Axin stability and consequently prevents Wnt/β-catenin signaling[103]BADR94, R96Prevents BAD phosphorylation by Akt and subsequent survival signaling[104]CaMKIIR9, R275Prevents CaMKII-dependent signaling in cardiomyocytes[105]CDK4R55, R73, R82, R163Prevents the formation of a CDK4/Cyc D3 complex and subsequent cell cycle progression[106]cTnIR146, R148Induces cardiac cell hypertrophy[107]EGFRR198, R200Upregulates EGFR signaling[108]ERαR260Promotes the formation of the ERα/PI3K/Src/FAK complex and subsequent activation of downstream kinase cascades[109]INCENPR887Enhances INCENP binding-affinity to AURKB and promotes cell division[110]KCNQR333, R345, R353, R435Promotes PIP2 binding and subsequent KCNQ channel activity[111]
MYCNR65Enhances MYCN stability through CDK-dependent phosphorylation[112]
NONOR251Favors NONO oncogenic function[113]
p38 MAPKR49, R149Promotes p38 MAPK phosphorylation by MKK3 and the subsequent activation of MAPKAK2 involved in erythroid differentiation[114]
Smad4R272Promotes Smad4 phosphorylation by GSK3 and support the activation of the canonical Wnt signaling[115]
Smad6R74, R81Participates in BMP signaling and prevents NF-κB activation[116,117]
Smad7R57, R67Facilitates TGF-β signaling[118]
TRAF6R88, R125Prevents TRAF6 ubiquitin ligase activity and regulates Toll-like receptor signaling[119]
TSC2R1457, R1459Blocks the Akt-dependent phosphorylation of TSC2 and regulates mTORC1 activity[120]

## 4. Cellular Features

### 4.1. Connection with Chromatin Dynamics and Transcriptional Regulation

Arginine methylation was first described as a PTM of histones that regulates reader protein recruitment and therefore chromatin dynamics. The main target of PRMT1 at the chromatin level is the arginine 3 of histone H4 (H4R3) [54,55]. Asymmetrically dimethylated H4R3, H4R3me2a, is associated with an active form of the chromatin and recognized by different Tudor domain-containing proteins, such as TDRD3 [121]. This protein, with no intrinsic activity, serves as a scaffold coregulator for the assembly of protein complexes at the transcription start sites of target genes. More precisely, TDRD3 can recruit, through its OB-fold domain, the DNA Topoisomerase IIIβ [122] and can directly interact with the RNA Polymerase II, previously methylated at R1810 by PRMT4 also known as CARM1 [123]. Therefore, this complex assembled through TDRD3 and likely involving other actors promotes transcription at H4R3me2a loci (Figure 2).

Interestingly, H4R3me2a can also recruit chromatin modifying enzymes involved in transcriptional regulation by depositing other histone marks on chromatin. Indeed, methylation of H4R3 by PRMT1 promotes the subsequent acetylation of H4K8 and H4K12 by the histone acetyltransferase p300 [56]. An H4R3me2a-dependent induction of H4K5 and H4K12 acetylation, allowing the recruitment of the transcription initiation factor TAFII250 and therefore contributing to chromatin opening, was also suggested using the chicken β-globin locus as a model [124]. Finally, the ability of H3R4me2a to act in trans to promote the acetylation of histone H3K9 and H3K14 by the histone acetyltransferases p300 and PCAF was demonstrated within the β-major globin promoter in murine erythroleukemia cells [124,125]. It is worth noting that PCAF directly interacts with H4R3me2a and this could explain how PRMT1-dependent methylation potentiates H3K9 and H3K14 acetylation [125] (Figure 2).

Conversely, the activity of PRMT1 on H4R3 is inhibited by the presence of acetylation, propionylation, crotonylation, butyrylation or 2-hydroxyisobutyrylation of H4K5 [126]. Moreover, H4K5ac combined with H4K8ac or H4K12ac increases its repressive effect on PRMT1 activity. There is currently one known exception, as acetylated H4K16 is associated with an increase in PRMT1 activity. Interestingly, the inducing effect of H4K16ac dominates the repressive effect of H4K5ac when the 2 histone marks co-exist [53,127] (Figure 2).

Aside from chromatin regulation, a large number of transcription factors whose activity can be regulated by PTMs are known PRMT1 substrates (Table 1). PRMT1-dependent methylation can notably increase their stability and thus promote their transactivation function. This type of mechanism has been described for FOXO1 whose methylation by PRMT1 prevents its proteosomal degradation and favors its nuclear localization [63]. The methyltransferase activity of PRMT1 can also impact interactions between transcription factors and their corepressors. For example, PRMT1 was shown to act as a coactivator of RUNX1 by inducing its methylation at R206 and R210, and thereby preventing its interaction with the transcriptional corepressor SIN3A [72]. Similarly, C/EBPα methylation at R35, R156 and R165 blocks its interaction with the corepressor HDAC3 [60]. 

### 4.2. Connection to Cell Signaling Pathways

#### 4.2.1. Steroid Receptors

To date, PRMT1 has been shown to methylate two steroid receptors; estrogen receptor (ERα) and progesterone receptor (PR). These arginine methylation events control different signaling pathways involved in breast tumorigenesis. 

##### Estrogen Receptor (ERα)

ERα regulates many physiological processes, notably the growth and survival of breast tumor cells, acting as a ligand-dependent transcription factor. Aside from the well described transcriptional effects, estrogen also mediates extranuclear events called non-genomic signaling via its receptor [128]. Our group showed that ERα is methylated on the residue R260 (met260ERα) by PRMT1 in response to estrogen or IGF-1 [109,129]. This event is a prerequisite for the formation of a signaling complex containing met260ERα, Src and PI3K, which orchestrates cell proliferation and survival. The involvement of this complex in breast carcinogenesis will be addressed in Section 5.1 of this review. Met260ERα is a transient event downregulated by the arginine demethylase JMJD6 [130].

###### Progesterone Receptor (PR)

Our group also demonstrated that PRMT1 methylates PR on the residue R637, within a RGG consensus site. This methylation event decreases PR stability in order to accelerate its recycling and its transcriptional activity. In addition, PRMT1 depletion decreases the expression of a specific subset of progesterone-target genes, involved in breast cancer cell proliferation and migration [68].

#### 4.2.2. Akt Signaling Pathway

Several reports demonstrated that specific arginine methylation, catalyzed by PRMT1 within the Akt consensus phosphorylation motif, works as an inhibitor of Akt-dependent survival signaling. 

##### FOXO

Forkhead box O (FOXO) is a family of transcription factors controlling a large variety of biological processes including cell survival [131]. Several studies revealed that FOXO proteins are phosphorylated by Akt, resulting (i) in the export of FOXO proteins from the nucleus to the cytoplasm [132,133] and (ii) in FOXO proteasomal degradation through polyubiquitination [134,135]. Interestingly, a member of the FOXO family, FOXO1 was shown to be methylated by PRMT1 on R248 and R250, in the consensus Akt phosphorylation site, impeding Akt phosphorylation on S253 [63]. This methylation event results in a decrease in its cytoplasmic localization and its subsequent degradation. PRMT1 depletion decreases oxidative-stress-induced apoptosis regulated by the Akt-FOXO1 pathway. These results indicated that PRMT1 arginine methylation can act as a modulator of Akt-phosphorylation by regulating responses to oxidative stress in mammalian cells.

##### BAD

Similarly, PRMT1 binds and methylates the proapoptotic protein BCL-2 antagonist of cell death (BAD) on R94 and R96, in the Akt consensus site. PRMT1 methylation on these two residues inhibits Akt phosphorylation on S99, a modification that is necessary for its interaction and sequestration with 14-3-3 proteins, resulting in cell survival [104]. 

#### 4.2.3. NF-κB Signaling

NF-κB plays an important role in the transcriptional regulation of genes involved in inflammation and cell survival. Toll-like receptor (TLR), when activated by lipopolysaccharides, triggers the recruitment of the adaptor protein Myd88 and the subsequent activation of the transcription factor NF-κB. TGFβ inhibits TLR signaling through the methylation of SMAD6 by PRMT1. Indeed, the binding of methylated SMAD6 to Myd88 results in its degradation, impeding TLR signaling to NF-κB [117]. Moreover, PRMT1 serves as a coactivator of NF-κB, synergistically with CARM1, although the underlying mechanisms are not fully elucidated [136]. More recently, the methylation of the RelA subunit of NF-κB by PRMT1 was identified as a repressive mark modulating TNFα/NF-κB response [70]. 

#### 4.2.4. Wnt Signaling

Wnt signaling plays important roles in embryonic development and cell proliferation. Aberrant Wnt signaling leads to several human diseases including cancer. Axin is a negative regulator of the Wnt pathway, as it is a key scaffold protein for the β-catenin destruction complex. PRMT1-induced methylation of axin enhances its interaction with GSK3β, leading to a decrease in axin ubiquitination and degradation [103]. Therefore, PRMT1 seems to be a new modulator of Wnt/β-catenin signaling. Moreover, PRMT1 also regulates this pathway by methylating substrates prior to their phosphorylation by GSK3β and its sequestration in endolysosomes, a key event in Wnt signaling [115]. Altogether PRMT1 appears as an important modulator of the Wnt pathway at the interface of protein phosphorylation and trafficking.

### 4.3. Cellular Role and Functions

#### 4.3.1. Embryogenesis and Development

The critical role of PRMT1 in embryogenesis and development was first suggested by the study of Pawlak et al. which showed that PRMT1 knockout mouse embryos, generated by insertion of a gene trap retrovirus in the second intron of the *PRMT1* gene, failed to develop beyond embryonic day 6.5, which would coincide with the exhaustion of the maternal stock of PRMT1 enzymes and methylated substrates [137]. It is worth noting that homozygous PRMT1 mutant embryonic stem (ES) cells isolated from mutant preimplantation blastocysts at day 3.5 are viable and retained the morphology and the same doubling time as wild-type ES cells. Moreover, in these cells, loss of PRMT1 activity is not balanced by the activation of other methyltransferases. Therefore, PRMT1 activity does not seem to be required for cell viability [137].

Early lethality of homozygous PRMT1 KO mouse embryos, as well as their uterus-enclosed localization, makes it difficult to study the epigenetic regulation of vertebrate development and emphasizes the importance to develop other models. Among them, Zebra fish embryos constitute a promising model as they are suitable for genetic manipulation approaches and express a highly conserved PRMT1 protein (90% identity with human PRMT1) at different stages of embryogenesis. A study conducted by Tsai et al. showed that PRMT1 knockdown, by antisense morpholino oligo injection into one-cell stage zebra fish embryos, induces developmental defects at gastrulation notably including a shortened body-length. This highlighted the importance of the methyltransferase activity of PRMT1 in early embryogenesis [138]. More recently, Shibata et al. used the TALEN genome editing technology to knockout PRMT1 in the diploid anuran *Xenopus tropicalis* that undergoes an external and biphasic development (embryogenesis and metamorphosis). They observed that H4R3me2a methylation by PRMT1 is not required for early embryogenesis but is essential for the growth and development of various organs including the brain, liver and intestine during late embryonic developmental stages, occurring prior to metamorphosis. This effect is directly related to the drastic inhibition of cell proliferation associated with PRMT1 KO in this model [139]. Interestingly, *Xenopus* embryos were already used to demonstrate the involvement of the xPRMT1b gene in early neural determination [140]. 

Another interesting aspect is the potential involvement of PRMT1 in placental development. A study of Sato et al. showed that murine placental expression of two PRMT1 isoforms is differentially regulated during the gestational period. More precisely, while PRMT1-v1 expression reaches a maximum at embryonic day E11 before decreasing, PRMT1-v2 expression increases from E13. This balance between the two isoforms explains the change in subcellular localization of PRMT1 observed between early and late stages of gestation; though further studies are required to determine the exact role played by PRMT1 in the placenta [141].

#### 4.3.2. DNA Damage Repair

The conditional knockout of PRMT1 in mouse embryonic fibroblasts is associated with a severe genetic instability characterized by the occurrence of spontaneous DNA damage, chromosome copy number variations and defective mitotic checkpoint [142]. The relevance of PRMT1 in the maintenance of genome integrity is based on the methylation and subsequent regulation of key factors involved in the major DNA repair pathways. 

The first substrate of PRMT1, involved in DNA damage repair, to be identified was MRE11 (Meiotic recombination 11). This component of the MRN complex (MRE11/RAD50/NBS1), recruited early upon DNA double-strand break (DSB), participates in the initiation of DNA repair pathways by homologous recombination (HR) or by non-homologous end joining (NHEJ). Methylation of the C-terminal GAR motif of MRE11 at R587 by PRMT1 does not seem to participate in the formation of the MRN complex but it promotes the relocalization of MRE11 from PML nuclear bodies to DNA-damage sites and it favors its exonuclease activity [92,99,100]. These events are essential to allow the recruitment of RAD51 and the subsequent activation of HR [100]. By using a model of knock-in mice that express the mutated MRE11RK protein devoid of methylarginines, Yu et al. also demonstrated that MRE11 methylation participates in the activation of the ATR/CHK1 checkpoint signaling [143]. Finally, methylated MRE11 is involved in telomere maintenance and regulates DNA replication by controlling the intra-S phase checkpoint in response to DNA damage [99,144]. 

The choice of pathways between NHEJ or HR is directly influenced by the DNA-end structure of DNA DSBs. Among the actors that play a pivotal role to orient this choice are the tumor suppressor protein BRCA1, which promotes HR repair by activating DNA-end resection, and p53-Binding Protein 1 (53BP1) that inversely activates NHEJ repair by inhibiting the recruitment of BRCA1 to DNA DSBs [145]. Interestingly, these two proteins are methylated by PRMT1, suggesting that arginine methylation may play an important role in directing the switch from HR to NHEJ repair pathways. More precisely, 53BP1 is methylated by PRMT1 at a canonical GAR motif localized in its kinetochore-binding domain and this methylation is essential for its DNA-binding activities [92,146]. Concerning BRCA1, the methylation status of the 504–802 protein region, that encompasses the DNA-binding domain, directly influences its interaction with transcription factors such as Sp1 or STAT1 and its subsequent recruitment to specific promoters [59]. 

The base excision repair mechanism (BER) that can correct single-stranded DNA breaks and oxidative or alkylation damage is also regulated by PRMT1, which methylates two major players in this pathway, namely the Flap endonuclease 1 (FEN1) and the DNA polymerase β (DNA Pol β). Methylation of FEN1 by PRMT1, at an arginine residue that remains to be determined, stabilizes the protein without disturbing its localization [96]. Moreover, unlike PRMT5-dependent methylation at residue R192 which strengthens the interaction between FEN1 and the DNA polymerase processivity factor PCNA necessary for a faithful and efficient BER, PRMT1-dependent methylation of FEN1 does not seem to impact this interaction [96,147]. Interestingly, methylation of the DNA Pol β by PRMT1 on R137 abolishes its binding with PCNA without affecting its enzymatic activities (polymerase and dRA-lyase) [94]. This suggests that methylation could regulate the sequential interaction of FEN1 and DNA Pol β with PCNA during BER. 

## 5. PRMT1 in Cancer

Since the substrates methylated by most PRMTs regulate various biological functions essential for cellular homeostasis, it is not surprising that a dysregulation of arginine methylation may contribute to cancer initiation and progression. The involvement of PRMT1 in carcinogenesis is no longer questioned due to its overexpression or aberrant splicing observed in numerous types of cancers. 

### 5.1. Breast Cancer

Various studies have shown that *PRMT1* gene expression is higher in breast tumor samples than in healthy tissue suggesting the involvement of PRMT1 in breast carcinogenesis [5,148]. Despite the detection of PRMT1-v1, v2 and v3 isoforms in breast tumor tissue, it seems that only the predominant PRMT1-v1 variant is correlated with clinical parameters such as histological grade [148].

ERα is an important PRMT1 substrate whose methylation can be associated with the development of breast cancer. Our group highlighted that a PRMT1-dependent hypermethylation of ERα at R260, induced in response to estrogen or IGF-1, is observed in different subtypes of breast cancers and regulates cell proliferation and survival [109,129]. We notably showed that the signaling complex containing met260ERα, Src and PI3K (described in Section 4.2.1 of this review) is expressed at low levels in the cytoplasm of normal mammary epithelial cells but highly expressed in 55% of breast tumors [149]. Moreover, its overexpression is correlated with the activation of Akt (pAkt), the main effector of the pathway, showing that this signaling pathway exists in vivo. In addition, a high expression of the complex is an independent marker of poor prognostic [149] and has been linked with resistance to tamoxifen [150,151].

Another interesting aspect is the key role of PRMT1 in the maintenance of stem-cell-like properties of breast cancer cells. PRMT1-dependent EGFR methylation on R198 and R200 upregulates different signaling cascades, notably those involving Akt, ERK or STAT3 in triple-negative breast cancer (TNBC) cells, MDA-MB-468. EGFR/ERK-dependent activation of ZEB1, a transcription factor that regulates epithelial-mesenchymal transition, may be implicated in cancer stem cell maintenance [152]. Interestingly, asymmetric dimethylation of H4R3 by PRMT1 at the ZEB1 promoter is another mechanism described to activate this factor and therefore promotes migration, invasion and acquisition of stem cell characteristics. It is worth noting that ZEB1 may simultaneously contribute to the PRMT1-dependent inhibition of senescence in breast cancer cells [153]. 

PRMT1-dependent methylation also inhibits the tumor suppressive function of some substrates. For example, methylation of C/EBPα at R35, R156 and R165 by PRMT1 prevents its interaction with the corepressor HDAC3, thus promoting the expression of cell-cycle genes such as cyclin D1 and the subsequent growth of breast cancer cells [60]. In the same line, BRCA1 methylation by PRMT1 affects its recruitment to responsive promoters but also its ability to interact with certain partners such as Sp1 or STAT1. As a result, this can significantly affect the tumor suppressive activity of BRCA1 [59].

### 5.2. Colorectal Cancer

Two clinical reports demonstrated the unfavorable prognosis associated with PRMT1 expression in colorectal cancer (CRC) patients by discussing the respective involvement of PRMT1-v1 and PRMT-v2 isoforms [154,155]. Mechanistically, it was described that H4R3me2a can recruit SMARCA4, an ATPase subunit of the SWI/SNF complex, to the promoter of certain target genes including EGFR to promote their expression. PRMT1-dependent enhancing of EGFR signaling is associated with a significant increase in the proliferative and migratory abilities of human CRC cells [156]. Moreover, methylation of EGFR at R198 and R200 by PRMT1 leads to an EGF-dependent hyperactivation of EGFR signaling and confers cells with resistance to the anti-EGFR monoclonal antibody, cetuximab. Indeed, in CRC patients, the rate of EGFR methylation is directly correlated with a higher recurrence rate after cetuximab treatment and a poorer overall patient survival [108].

Recently, the non-POU domain-containing octamer-binding protein (NONO), which is overexpressed in CRC tissue, was described as a substrate of PRMT1. Methylation of NONO at R251 is required to promote its oncogenic function including the induction of CRC cell proliferation, migration and invasion [113].

### 5.3. Lung Cancer

As described for other cancers, PRMT1 expression is significantly increased in lung cancer tissue compared to non-neoplastic ones though very little data are available in the literature to explain its role in lung carcinogenesis [157]. A study by Avasarala et al. highlighted that PRMT1 participates in non-small cell lung cancer progression and metastasis through the methylation of the EMT-associated transcription factor Twist1 at R34. PRMT1-dependent Twist1 methylation is associated with inhibition of E-cadherin expression [78]. Moreover, PRMT1 can methylate the inner centromere protein (INCENP) at R887 to favor its interaction and the subsequent activation of aurora kinase B in A549 non-small cell lung cancer cells. This mechanism regulates the alignment and segregation of chromosomes during cell division to promote the growth of cancer cells [110]. 

### 5.4. Other Cancers

Dysregulation of PRMT1 expression has been reported in several other types of cancers, albeit the molecular mechanisms that drive the initiation and progression of these cancers remain incompletely understood. The limited data available in the literature indicate that PRMT1 is particularly dysregulated in bladder cancer, esophageal squamous cell carcinoma, as well as in acute myeloid leukemia [157,158,159]. Interestingly, in ovarian carcinomas, upregulation of PRMT1 expression is associated with an increased methylation of the apoptosis signal-regulated kinase 1 (ASK1), which confers tumor cells with resistance to platinum-based chemotherapeutic agents [160]. Moreover, in prostate cancer, the methylation status of H4R3 is significantly correlated with clinical features, such as tumor grade or the risk of prostate cancer recurrence. This study highlighted the fact that histone modifications can also serve as a prognostic marker [161].

### 5.5. PRMT1 Inhibitors

In 2004, the symmetrical sulfonated urea salt named arginine methylation inhibitor-1 (AMI-1) was the first PRMT inhibitor characterized [162]. Since then, two substrate competitive inhibitors, MS023 and GSK3368715, that broadly target type I PRMTs (Table 2), were developed and displayed antitumor activities notably on xenograft mouse models of acute myeloid leukemia or breast cancer, respectively [163,164,165]. Promisingly, the GSK3368715 inhibitor is currently undergoing a first-time clinical trial (NCT03666988) for patients with solid tumors and diffuse large B-cell lymphoma. However, high affinity of these inhibitors for other type I PRMTs, renders the identification and characterization of specific PRMT1-dependent effects difficult. 

Currently, two PRMT1-specific inhibitors, TC-E-5003 and C7280948, are mentioned in the literature (Table 2). TC-E-5003 displays significant antitumor activity in vitro on breast or lung cancer cell lines and inhibits the growth of xenografted A549 lung cancer cells in mice [166]. Concerning C7280948, a study of Yin et al. showed that it suppresses colorectal cancer cell proliferation, migration and invasion [113]. Additionally, a structure-based virtual screening of different libraries of compounds allowed the identification of several potential PRMT1-specific inhibitors, the properties of which were detailed by Hu et al. [167]. Although these inhibitors are promising, more studies are needed to characterize and consider their clinical potential. 

## 6. Outlook

Over the last twenty years since the discovery of PRMT1, the number of studies conducted on this enzyme has constantly increased. This interest, which persists today, has improved our knowledge on the diversity of its substrates and the numerous biological functions regulated by PRMT1. Its key role in cancer initiation and progression makes PRMT1 an interesting target for the development of new anti-cancer therapeutic strategies. Therefore, the development of inhibitors that target PRMT1 activity is an ongoing challenge that may offer new therapeutic opportunities for various pathologies in the coming years.

## Figures and Tables

**Figure 1 life-11-01147-f001:**
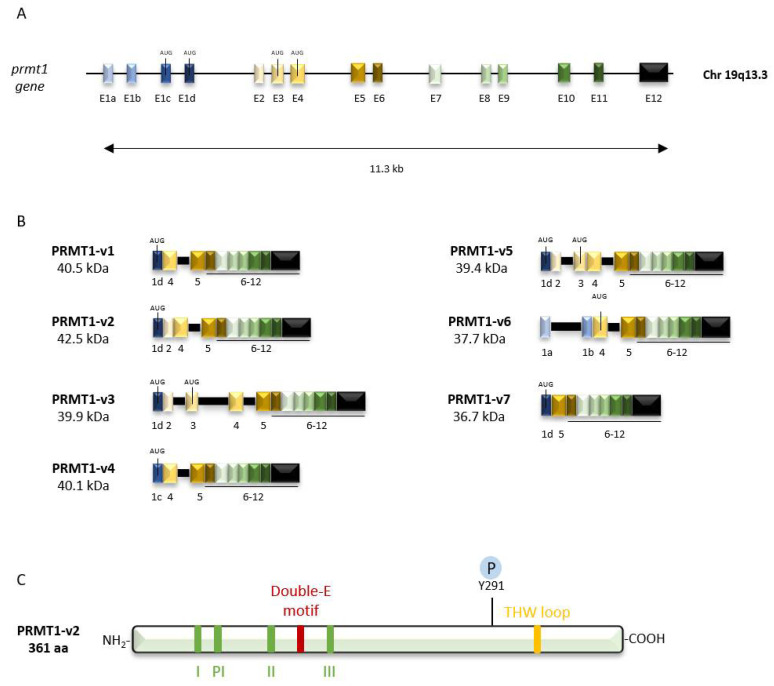
Genomic organization and protein structure of PRMT1. (**A**) Genomic organization of the *PRMT1* gene which spans 11.3 kilobases (kb) and possesses 12 constitutive exons including exon 1 subdivided into 4 alternative exons, represented by a scale of blue-colored boxes; (**B**) Exon composition of the different PRMT-1 isoforms (v1 to v7). The sequences of intron boundaries are represented by the black boxes. Molecular weight of each protein isoform is indicated in kilodaltons (kDa); (**C**) Protein structure of PRMT1-v2. The PRMT signature motifs (I, Post-I (PI), II, III) as well as double-E motif, the THW loop and the phosphorylation site Y291 are represented (adapted from [5,6]).

**Figure 2 life-11-01147-f002:**
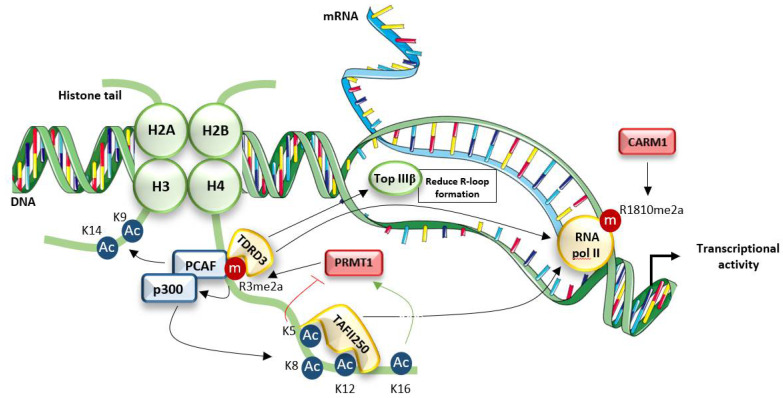
PRMT1 regulates chromatin dynamics. PRMT1-dependent H4R3 methylation (R3me2a) allows the recruitment of the Tudor domain-containing protein, TDRD3, which in turn associates with topoisomerase IIIβ (Top IIIβ) to reduce R-loop formation and RNA polymerase II (RNA pol II) to promote transcriptional activity. Concomitantly, H4R3me2a-dependent activation of histone acetyltransferases p300 and pCAF induces acetylation of H4K5, H4K8, H4K12 but also of H3K9 and H3K14. H4K5 and H4K12 are involved in the recruitment of TAFII250 that associates with RNA pol II. H4K5ac and H4K16ac are also involved in PRMT1-activity regulation. Ac = Acetylation, m = methylation.

**Table 2 life-11-01147-t002:** List of PRMT inhibitors targeting PRMT1. ND: Not defined in literature.

Name	Mechanism of Action	Target(s)	IC50	Reference
AMI-1	Substrate competitive SAM uncompetitive	PRMT1	8.81 µM	[162]
MS023	Substrate competitive SAM uncompetitive	PRMT1	30 nM	[163]
		PRMT3	119 nM	
		PRMT4/CARM1	83 nM	
		PRMT6	4 nM	
		PRMT8	5 nM	
GSK3368715	Substrate competitive SAM uncompetitive Reversible	PRMT1	33.1 nM	[165]
		PRMT3	162 nM	
		PRMT4/CARM1	38 nM	
		PRMT6	4.7 nM	
		PRMT8	3.9 nM	
TC-E-5003	ND	PRMT1	1.5 µM	[166]
C7280948	Interaction with the substrate-binding pocket	PRMT1	12.8 µM	[113]

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
