# Peer review of "Structure, Activity, and Function of PRMT1"

_life, 2021, doi:10.3390/life11111147_

Round 1

Reviewer 1 Report

Thiebaut et al. nicely summarise current knowledge on the structure, function and role in cancer of the protein arginine methyltransferase PRMT1.

The review is well written, logically structured, and generally detailed in the areas it covers. One area that could be expanded on is the very limited coverage of available inhibitors for PRMT1, which are only briefly mentioned in the outlook section. I would also like to see the inclusion of a figure showing the crystal structure of PRMT1 (ideally highlighting the residues that are mentioned in the text) as structural features of PRMT1 are a main focus and would round off the review.

Overall though, this is an informative and thorough review of the current literature that many readers in the field will enjoy.

Author Response

Reviewer #1: Thiebaut et al. nicely summarise current knowledge on the structure, function and role in cancer of the protein arginine methyltransferase PRMT1.

The review is well written, logically structured, and generally detailed in the areas it covers. One area that could be expanded on is the very limited coverage of available inhibitors for PRMT1, which are only briefly mentioned in the outlook section. I would also like to see the inclusion of a figure showing the crystal structure of PRMT1 (ideally highlighting the residues that are mentioned in the text) as structural features of PRMT1 are a main focus and would round off the review.

Overall though, this is an informative and thorough review of the current literature that many readers in the field will enjoy.

Response :

We appreciate the interest shown by reviewer 2 in our work and we would like to thank him/her for his/her constructive comments.

  • We agree with the reviewer that the paragraph about PRMT1 inhibitors needed to be expanded in view of their promising clinical potential. Therefore, we have developed the discussion on this point in a new section “5.5 PRMT1 inhibitors” and added a summary table “Table 2. List of PRMT inhibitors targeting PRMT1” (Page 16 of 27).
  • We thank the reviewer for his/her relevant suggestion concerning the addition of a figure showing the crystal structure of PRMT1. An extensive study of the crystal structure of rat PRMT1 which shares 96% identity with the amino acid sequence of human PRMT1 was performed by Zhang and Cheng in 2003 and highlighted all the conserved domains and residues mentioned in the text. Therefore, we have added a sentence in the section 2.2 (Page 3 of 27) to refer the reader to this very detailed and informative study about the structure of PRMT1.

Reviewer 2 Report

With great pleasure, I have read the article entitled: Structure, Activity, and Function of PRMT1. This unit has been recognized as the main arginine methyltransferase enzyme in mammalian cells. This protein is important for several endocellular processes: regulation of chromatin dynamics and transcription, cell signalling pathways, DNA lesion repair. As a consequence of the above PRMT1 can play a significant role in cancer initiation and progression. Therefore, it looks as a suitable target for anticancer therapy, which has been denoted by the authors in this article.  From the editorial point, the article is well organized, focuses step by step on the “story” of PRMT1 protein. Moreover, the article is well written and readable.

Due to the above, I recommend this manuscript for publication in the Life journal.

Author Response

Reviewer #2: With great pleasure, I have read the article entitled: Structure, Activity, and Function of PRMT1. This unit has been recognized as the main arginine methyltransferase enzyme in mammalian cells. This protein is important for several endocellular processes: regulation of chromatin dynamics and transcription, cell signalling pathways, DNA lesion repair. As a consequence of the above PRMT1 can play a significant role in cancer initiation and progression. Therefore, it looks as a suitable target for anticancer therapy, which has been denoted by the authors in this article.  From the editorial point, the article is well organized, focuses step by step on the “story” of PRMT1 protein. Moreover, the article is well written and readable.

Due to the above, I recommend this manuscript for publication in the Life journal.

Response :

We kindly thank Reviewer 2 for his/her great comments about our work.